# Steel Surface Doped with Nb via Modulated Electron-Beam Irradiation: Structure and Properties

Maxim Sergeevich Vorobyov, Elizaveta Alekseevna Petrikova, Vladislav Igorevich Shin, Pavel Vladimirovich Moskvin, Yurii Fedorovich Ivanov, Nikolay Nikolaevich Koval, Tamara Vasil'evna Koval, Nikita Andreevich Prokopenko, Ruslan Aleksandrovich Kartavtsov and Dmitry Alekseevich Shpanov *

Institute of High Current Electronics, Siberian Branch, Russian Academy of Sciences, 2/3 Akademichesky Ave., Tomsk 634055, Russia; vorobyovms@yandex.ru (M.S.V.); elizmarkova@yahoo.com (E.A.P.); shin.v.i@yandex.ru (V.I.S.); pavelmoskvin@mail.ru (P.V.M.); yufi55@mail.ru (Y.F.I.); koval@opee.hcei.tsc.ru (N.N.K.); tvkoval@tpu.ru (T.V.K.); nick08_phantom@mail.ru (N.A.P.); kartavcov@gmail.com (R.A.K.)
* Correspondence: das138@tpu.ru

**Abstract:** A niobium film on an AISI 5135 steel substrate was exposed to submillisecond pulsed electron-beam irradiation with controlled energy modulation within a pulse to increase the film–substrate adhesion. This modulated irradiation made it possible to dope the steel-surface layer with Nb through film dissolution in the layer, for which optimum irradiation conditions were chosen from experiments and a mathematical simulation. The irradiated system was tested for surface hardness and wear, and its surface structure and elemental composition were analyzed. The results demonstrate that the microhardness of the irradiated system is much higher and that its wear rate is much lower compared to the initial state.

**Keywords:** surface modification; doping; grid plasma cathode; electron beam; Nb–steel system; high-rate crystallization





## 1. Introduction

Currently, structural steels occupy one of the leading positions in industry due to their strong mechanical characteristics, good technological efficiency, and comparatively low cost [1–3]. In particular, they are widely used to manufacture products used in operations with high loads, and because such loads can cause deformation and fracturing (e.g., cracking, spallation [4]), mostly in their surface layers, the surface layers of structural steels are hardened [5]. One of the ways of hardening the surfaces and improving the functional properties of machine parts is to deposit a protective coating on their working surfaces, which can greatly improve their physicomechanical characteristics (strength, wear resistance, etc.) and service life [6]. However, such protectively hardened coatings often suffer from weak adhesion to substrate materials. This problem can be solved by combining several hardening methods so that each shows its advantages and not its disadvantages [7]. The formation of high-strength and thermally stable surface layers through refractory-metal-coating deposition and intense pulsed electron-beam irradiation with amplitude and width modulation makes it possible to greatly improve the service conditions and applicability of structural steels [8–10]. Structural steels doped with Nb, compared to other steels, are lighter and harder, and they have greater strength, higher corrosion resistance, and longer lifetimes; additionally, they are ductile, i.e., they show no brittleness after doping with Nb. The fields in which they are used include mechanical, radio, and nuclear energy engineering, chemical industries, the automobile and building industries, and the aerospace industry in particular [11–22]. Steel doping with V, Ti, Nb, and Zr yields carbides that are poorly soluble in austenite. These elements are efficient (in decreasing the grain size, cold-brittleness threshold, and sensitivity to stress concentration) only if their steel

content is low (up to 0.15 wt.%). With higher contents, they decrease the hardenability and brittle-fracture resistance of steel because a large amount of metal carbides precipitates at the grain boundaries. The presence of Nb in proper amounts in steels provides primary structural refinement due to ferrite formation and the filling of its interdendrite spacings with eutectic liquid. With these positive effects, for example, low-alloy structural AISI 5135 steel doped with Nb can be used to manufacture tools for the efficient extrusion of light metals (e.g., Al) and the hot working of bearings; thus, it can be a good alternative to expensive, difficult-to-machine heat-resistant steels doped in their bulk with Mo, W, Ti, and V. The surface modification considered below is a form of finishing and does not require any further treatment [23–27].

## 2. Test Material and Treatment Methods

The test material was low-carbon AISI 5135 steel ((0.1–0.39) C, (0.17–0.37) Si, (0.5–0.8) Mn, $\leq$0.3 Ni, $\leq$0.035 Cr, $\leq$0.035 P, 0.8–1.1 Cr, $\leq$0.3 Cu, and balance Fe, wt.%). Its specimens, with dimensions 15 $\times$ 15 $\times$ 5 mm, were coated with an Nb film 3 μm thick via plasma-assisted arc deposition on the QUINTA vacuum ion-plasma setup [28], which is part of the UNIKUUM complex of unique electrophysical equipment of Russia (https://ckp-rf.ru/catalog/usu/434216/ (accessed on 13 June 2023)). Figure 1 shows a schematic diagram of Nb-film deposition. The deposition was realized using a PINK-P gas plasma generator based on a non-self-sustained low-pressure arc discharge with a hollow and a hot cathode (Figure 1, cathodes 4 and 5, respectively). The generator makes it possible to treat objects up to 40 cm in length and up to 1 kg in weight. It is used for preliminary surface cleaning and heating in argon plasma and for ion-plasma assistance in vacuum-arc deposition, which increases the coating–substrate adhesion and the vacuum-arc stability. The arc evaporator with a 98 wt.% Nb cathode (Figure 1, evaporator 2), compared to its previous version, ensured better cooling of the cathode back's surface and a smaller amount of microdroplets in the coating. Before film deposition, the specimens were washed with petrol in an ultrasonic bath (to remove mechanical and oil contaminants from their surfaces) and, after they were fixed in the specimen holder, they were placed in the vacuum chamber at a distance of 17 cm from the evaporated cathode. Subsequently, the vacuum chamber was pumped to a limiting pressure of $1 \times 10^{-2}$ Pa, and the specimens were coated with a Nb film. The deposition comprised several stages. First, the specimens were exposed to ion-plasma surface cleaning in argon. Subsequently, their surfaces were bombarded with Nb ions at an argon pressure of 0.15 Pa, arc current of 80 A, substrate-bias voltage of −900 V, and pulse duty factor of 85%, which ensured better film adhesion. Next, the specimens were coated with Nb for 20 min at an argon pressure of 0.3 Pa, arc current of 80 A, and bias voltage of 50 V. The Nb-coated specimens were cooled in the vacuum chamber to room temperature and removed from it for further pulsed electron-beam treatment. The coating thickness determined by the Calotest method was 3 μm.

The Nb film–AISI 5135 steel-substrate system was irradiated with an electron beam on the SOLO setup (also a part of the UNIKUUM complex), whose electron source allows the generation of electron beams with diameters of up to 5 cm, energy of up to 25 keV, and energy density of up to 100 J/cm$^2$ at a pulse duration of 20–1000 μs [29]. The unique feature of this type of electron source is the possibility of controlling the beam current, which is weakly dependent on the accelerating voltage [30]. Thus, it is possible to control the beam power in the submillisecond range of pulse durations [31] and, hence, the rate of energy delivery to the surface of a treated target within a beam current pulse [32]. The control of energy delivery makes it possible to control the temperature field in the target surface layer and, hence, its structure and phase state.

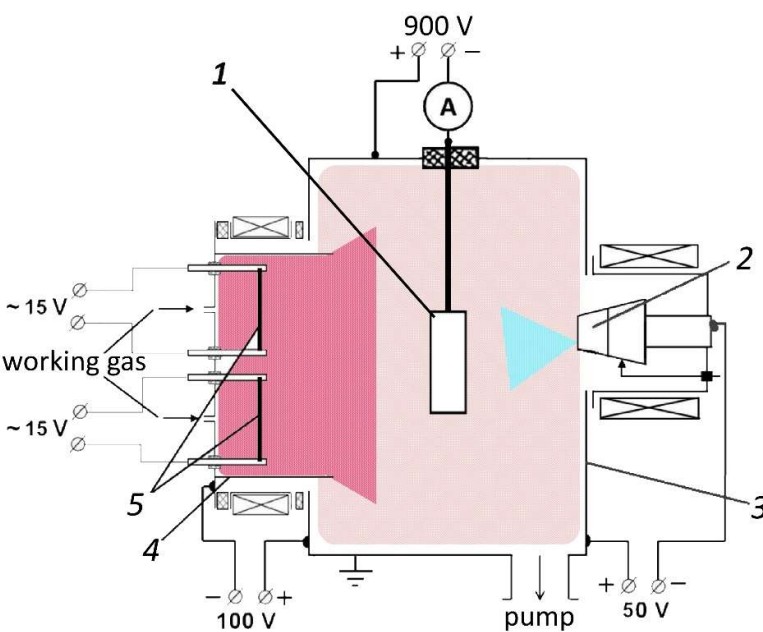

**Figure 1.** Schematic diagram of Nb-film deposition: 1—specimen holder, 2—arc evaporator with Nb cathode, 3—vacuum chamber (anode), 4—hollow cathode of PINK-P generator, 5—hot W cathode of PINK-P generator.

Figure 2 shows a schematic diagram of electron-beam irradiation. Niobium-coated steel specimen *8* was fixed on two-coordinate manipulator 6 with narrow stainless steel plate *7* with thickness 0.2 mm and height 5–7 mm for thermal-loss reduction. Using the manipulator table, the specimens were moved in vacuum chamber *3* under electron beam *5*. The temperature of the specimen surface was measured with high-speed infrared pyrometer *15* (Kleiber KGA 740-LO), allowing temperature measurements in the range of 300–2300 °C with a response time of 6 μs. The pyrometer used radiation at 2–2.2 μm, which was collected by lens *11* with a surface-spot diameter of about 5 mm at the specimen center. The output signal of the pyrometer was the voltage measuring 0–10 V, which depended linearly on the temperature in the operating range of 2000°. To determine the specimen-surface emissivity, the specimen was heated to 500 °C for 3 min by an electron beam for a duration of 200 μs and repetition frequency of 10 Hz, which did not change the surface properties. After this heating, the heat was distributed over the specimen volume via conduction, such that the specimen was cooled to 300 °C in about 2 min. During the period of cooling, the specimen-surface temperature was measured with the pyrometer, as was the temperature in the specimen bulk with K-type thermocouple *12* built in the specimen on the back side. The results of temperature measurements were compared to determine the surface emissivity.

The modes of irradiation were chosen so that the specimen's surface layer would be heated above the steel's melting temperature (1450–1550 °C) but below the niobium's melting temperature (2500 °C), i.e., to ≈2000 °C. Thus, it was expected that the thin Nb film would dissolve in the molten steel-surface layer kept at 2000 °C. According to the pyrometer specifications, the measured temperature $T$ [°C] is determined by its output signal $U_p$ [V] as follows: $T = 300 + 200\,U_p$. The behavior of $U_p$ with time for different irradiation modes is presented in Figure 3.

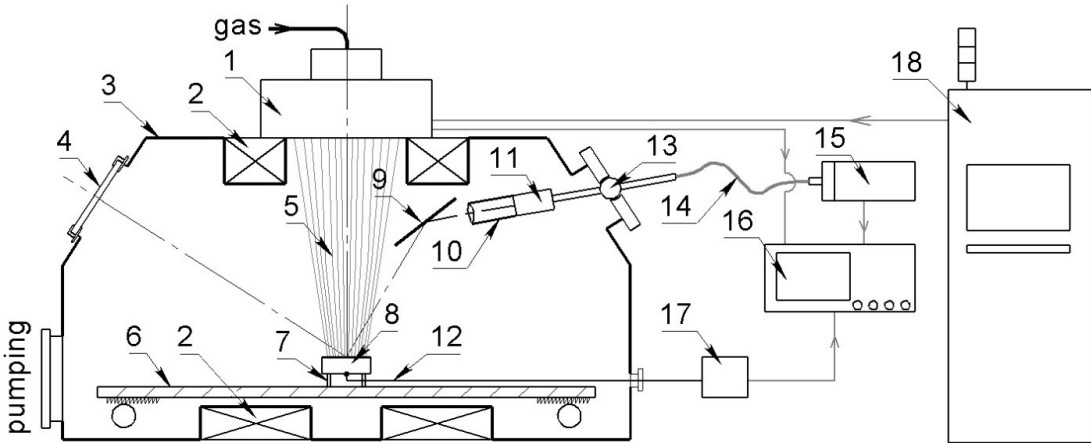

**Figure 2.** Schematic diagram of electron-beam irradiation: 1—electron source, 2—magnetic coils, 3—vacuum chamber, 4—observation window, 5—pulsed electron beam, 6—two-coordinate manipulator table, 7—fastening plates, 8—irradiated specimen, 9—copper mirror, 10—collimator, 11—lens, 12—thermocouple, 13—vacuum joint, 14—fiber-optic cable, 15—high-speed infrared pyrometer (Kleiber KGA 740-LO), 16—oscilloscope, 17—normalizing converter, 18—power-supply units and automation system.

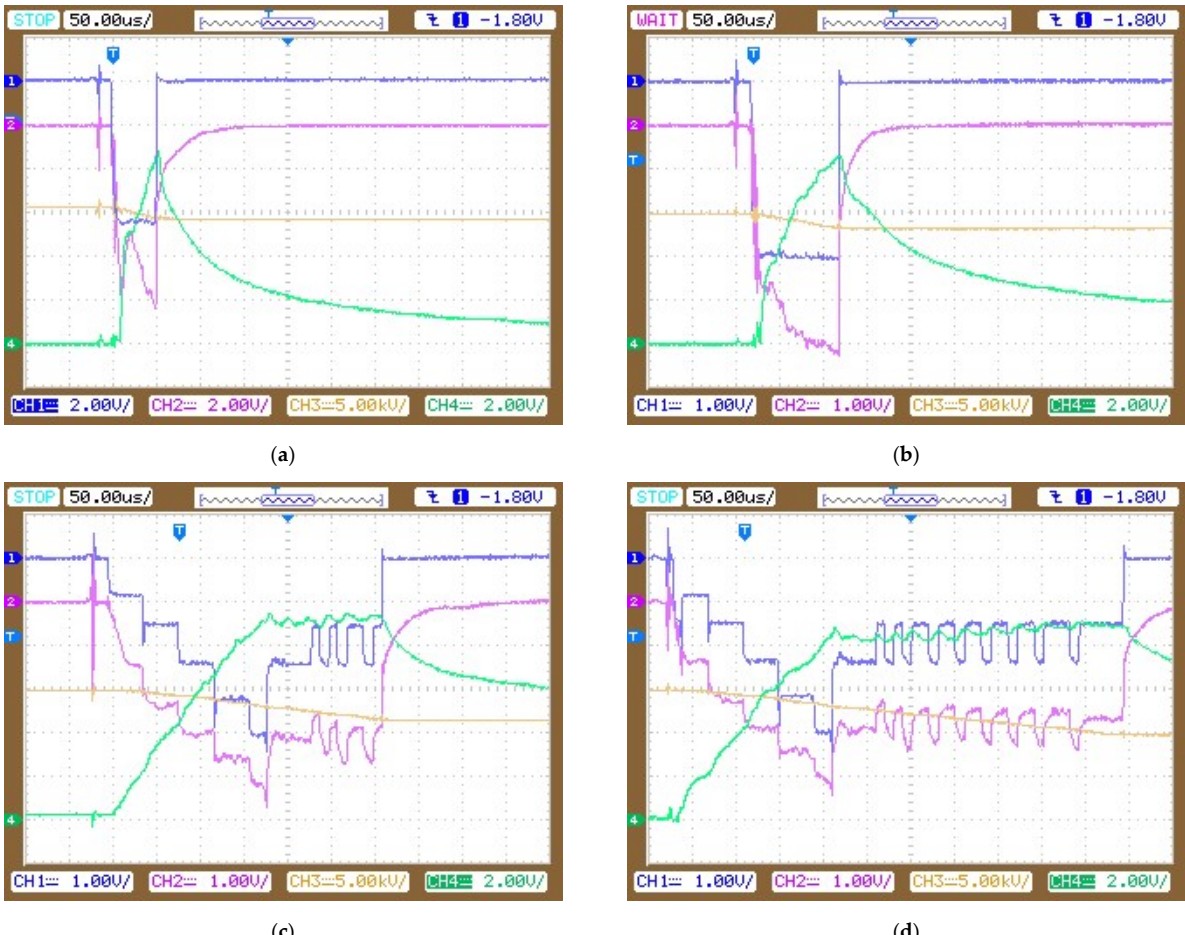

**Figure 3.** The behavior of Up with time for different irradiation modes: (**a**,**b**)—rectangular current pulses of duration 50 μs and 100 μs; (**c**,**d**)—pulses with varied beam currents at which the surface temperature reached 2000 °C and remained at this level for 150 μs for a total pulse duration of 300 μs and for 350 μs at 500 μs.

To maintain the specimen surface at the same temperature level, the following parameters were chosen for single beam current pulses:

- rectangular current pulses of duration 50 µs (Figure 3a) and 100 µs (Figure 3b);
- pulses with varied beam currents at which the surface temperature reached 2000 °C and remained at this level for 150 µs for a total pulse duration of 300 µs (Figure 3c) and for 350 µs at 500 µs (Figure 3d).

The following color symbols were used on the waveforms (Figure 3): waveforms of discharge current $I_d$ (1, blue, 40 A/div for figure "a"and 20 A/div for others), acceleration-gap current $I_0$ (2, magenta, 40 A/div for figure "a" and 20 A/div for others), accelerating voltage $U_0$ (3, brown, 5 kV/div, zero is the same as signal 4), and pyrometer signal $U_p$ (4, green, 2 V/div) for different irradiation modes. The beam was transported to the sample in the plasma that it created. Therefore, current $I_0$ contained not only the beam current, but also the ion current, which moved from the plasma to the high-voltage electrode.

Figure 4 shows experimental beam-power profiles for the beam-current pulses in Figure 3. It is shown that the beam-current modulation within a submillisecond pulse due to proportional arc-current modulation made it possible to control the beam power and the energy to the target surface within the pulse and, hence, the temperature field in the specimen-surface layer.

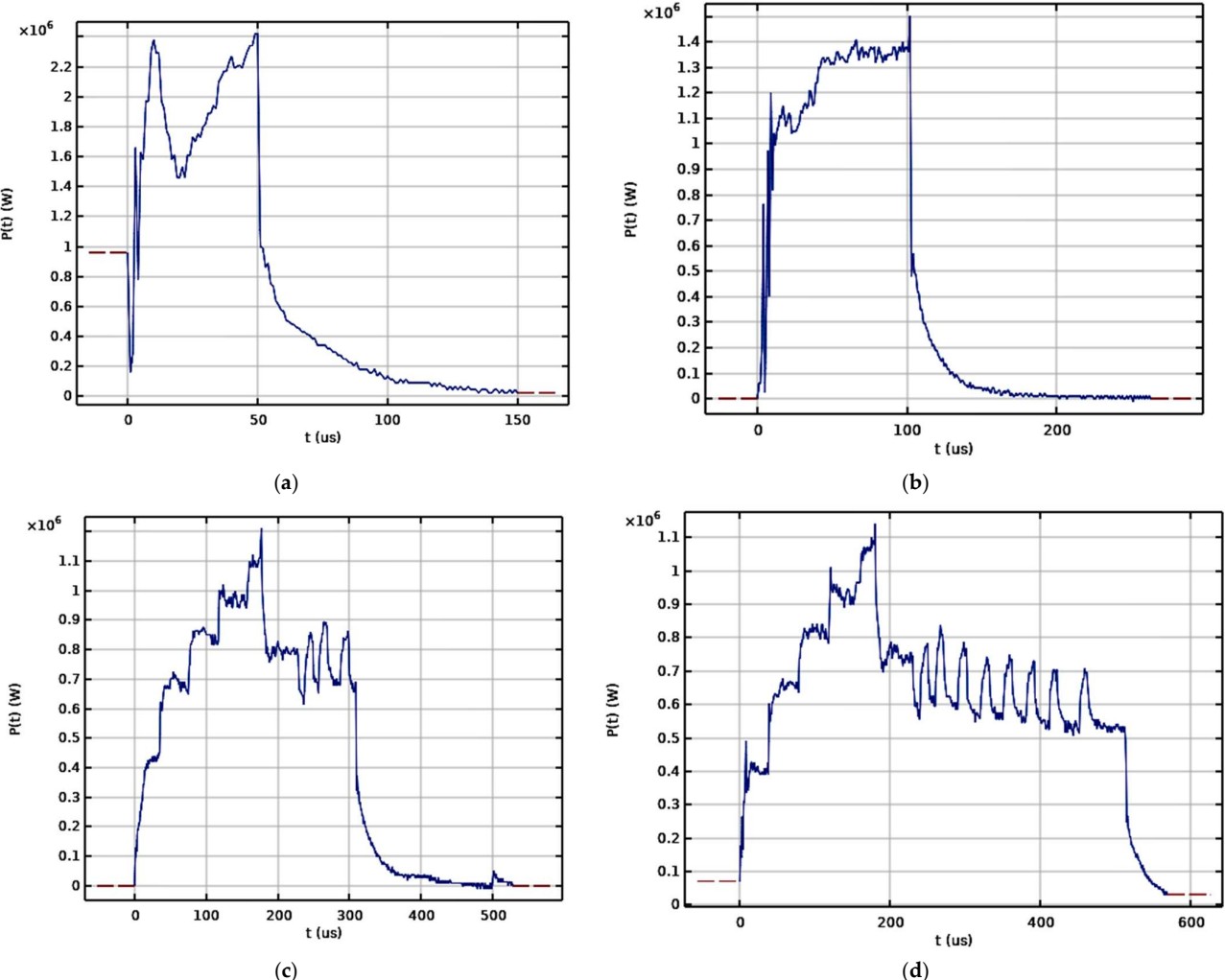

**Figure 4.** Experimental beam-power profiles within pulses lasting 50 (**a**), 100 (**b**), 300 (**c**), and 500 µs (**d**).

The temperature field formed in the Nb film–AISI 5135 steel-substrate system under intense pulsed electron-beam irradiation (beam energy density 10–50 J/cm$^2$, current pulse duration 50–500 µs) was estimated using a mathematical simulation.

With an average energy of the beam electrons on the target of ~10 keV, the energy source in the mathematical model can be assumed to act on the surface, and the thermal processes for pulse durations of 50–500 µs can be considered in the one-dimensional approximation because the transverse size of energy exposure is much larger than the depth of thermal-field propagation. With these assumptions, the estimation of the heating is reduced to solving the heat-conduction equation:

$$\rho c \frac{\partial T}{\partial t} = \frac{\partial}{\partial x}\left(\lambda \frac{\partial T}{\partial x}\right) \tag{1}$$

with a boundary and initial conditions of

$$-\lambda \frac{\partial T(t,0)}{\partial x} = q(t), \ \frac{\partial T(t,l)}{\partial x} = 0, \ T(t=0, \text{x}) = T_0 \tag{2}$$

where $c$ is the specific heat capacity, $\rho$ is the density, and $\lambda$ is the heat conductivity, which is dependent on the temperature and coordinates and corresponds to the coating or substrate, $q(t) = U(t)j(t)$ is the power density of an external heat source, $U(t)$ and $j(t)$ are the accelerating voltage and the beam-current density, respectively, $l$ is the computational domain length, $T_0 = 23\ °C$.

Formula (2) is a boundary condition of the second kind (Neumann's condition): a solid is heated from the outside by a stream of high-temperature thermal radiation $q(t)$ and the initial value.

The mathematical model assumes the presence of a two-phase region. In a solid–liquid system, this region is characterized by the average liquid volume fraction, $\theta$. The phase transition falls in the temperature interval, $\Delta T$, at which pint the material state is modeled by a smoothed function of $\theta$, varying from 1 to 0 [33]. The effective heat conductivity of the solid–liquid system $\lambda$ is related to the solid conductivity $\lambda_s$ and liquid conductivity $\lambda_l$, as

$$\lambda = (1-\theta)\lambda_s + \theta\lambda_l \tag{3}$$

The density of the two-phase region and its heat capacity were calculated in the same way. The latent heat of fusion $L$ is introduced as an additional term in the heat capacity in the phase transition: $c_i = c_s + L/\Delta T$. The value of the phase-transition interval $\Delta T$ is determined from the correspondence between the calculated and experimental surface temperatures.

The problem was solved numerically for table values of thermophysical parameters [34]. For AISI 5135 steel, the melting temperature is $T_l = 1420\ °C$, $\Delta T = 8\ °C$, and the heat conductivity and the heat capacity at $T < T_l$ measure $\lambda_s = 45.62 - 0.003T - 4.6 \times 10^{-5}T^2 + 3 \times 10^{-8}T^3$ W/(m·°C) and $c_s = 430.75 + 0.3802T - 0.00014T^2$ J/(kg·°C), $\lambda_l = 35$ W/(m·°C), $\rho = 7820$ kg/m$^3$, $L = 84$ kJ/kg, respectively. For the coating of thickness 3 µm, the Nb's melting temperature is 2741 °C, $\rho_s = 8000$ kg/m$^3$, $c_s = 268 + 0.048T$ J/(kg·°C), $\lambda_s = 53 + 0.013T$ W/(m·°C). The coating melting temperature is higher than the maximum surface temperature of 2000 °C considered in this study. The coating fails under electron-beam irradiation, changing its properties due to the molten substrate. This is allowed for in the simulation by estimating the thermophysical coefficients as averages for the coating and substrate.

For model pulses close to the experimental pulses (Figure 4), we numerically analyzed the temperature-field dynamics in the Nb film–AISI 5135 steel-substrate system on approaching the surface temperature of 2000 °C. For two rectangular pulses with durations of 50 µs and 100 µs and energy densities of 17 J/cm$^2$ and 24 J/cm$^2$, respectively, the surface temperature of 2000 °C was reached at the end of the pulses. The melt depths in these modes were 8 µs and 12 µs, respectively. To increase the thickness of the molten-steel layer,

calculations were performed in which the specimens surface was preliminarily heated to ≈200 °C for 200 µs and kept at this temperature for 100 µs and 300 µs (pulses with durations of 300 µs and 500 µs). Figure 5a shows the beam power and the surface temperature for the pulse of duration, 500 µs. The temperature distribution in depth from the surface at different points in time is presented in Figure 5b.

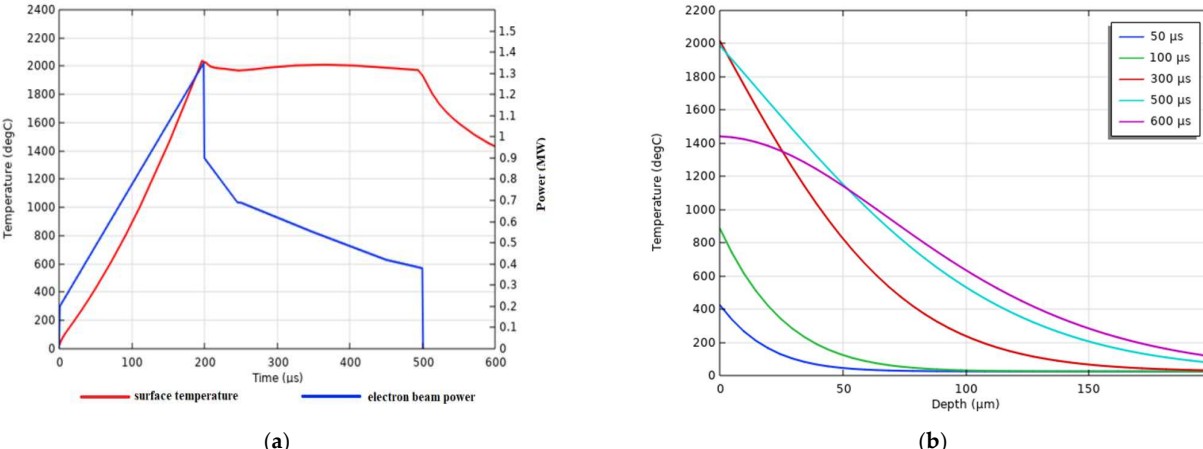

(**a**) (**b**)

**Figure 5.** Surface temperature and beam power vs. time (**a**), and temperature distribution in depth at different time points (**b**) for model pulse of duration 500 µs (for mode in Figure 4d).

In a real electron beam, as numerical calculations show, the following physical processes take place:

(1) energy dissipation (up to 20%) of electrons transported to the target in an extended plasma channel;

(2) a potential drop in the accelerating gap (up to 50%) due to the growth of the spatial charge with an increase in the beam current.

The maximum loss in the target is at a distance of one-third of the depth of the run, so the surface source in the mathematical model is quite correct at $Re \ll 3\sqrt{at}$, and deducted thermal conductivity.

The molten-layer depth depends both on the pulse duration and on the beam power density. Figure 6 shows the melt depths, and the surface melt times for the pulses considered. The melt depth at pulse durations of 50–100 µs was 8–12 µm, and at 500 µs, it increased to 30 µm (Figure 6a), with the surface remaining molten for 480 µs (Figure 6b).

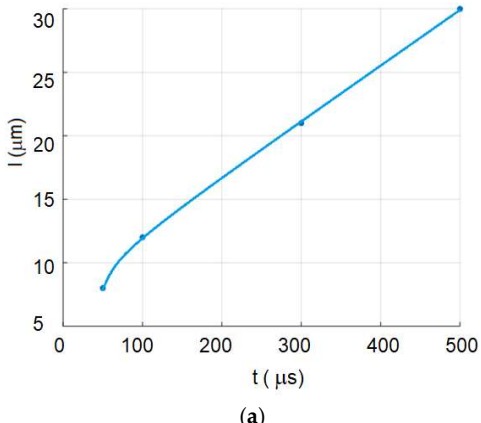
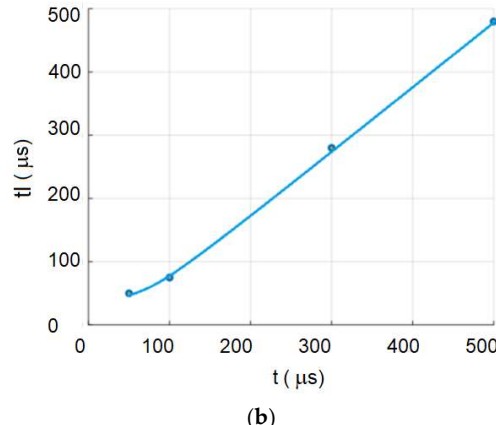

(**a**) (**b**)

**Figure 6.** Melt depths (**a**) and surface melt times (**b**) for model pulses with durations of 50, 100, 300, and 500 µs.

Using the data from numerical calculations, we chose the irradiation modes (Figure 4) that provided surface modifications, with Nb kept in its near-melting state and AISI 5135 steel in its surface-melting state.

Figure 7 shows the surface temperature $T(t)$ according to the calculation and experimental data for different beam powers and durations (Figure 4). The pulsed character of $T(t)$ with the surface temperature remaining constant was due to beam-current fluctuations (Figure 4c,d).

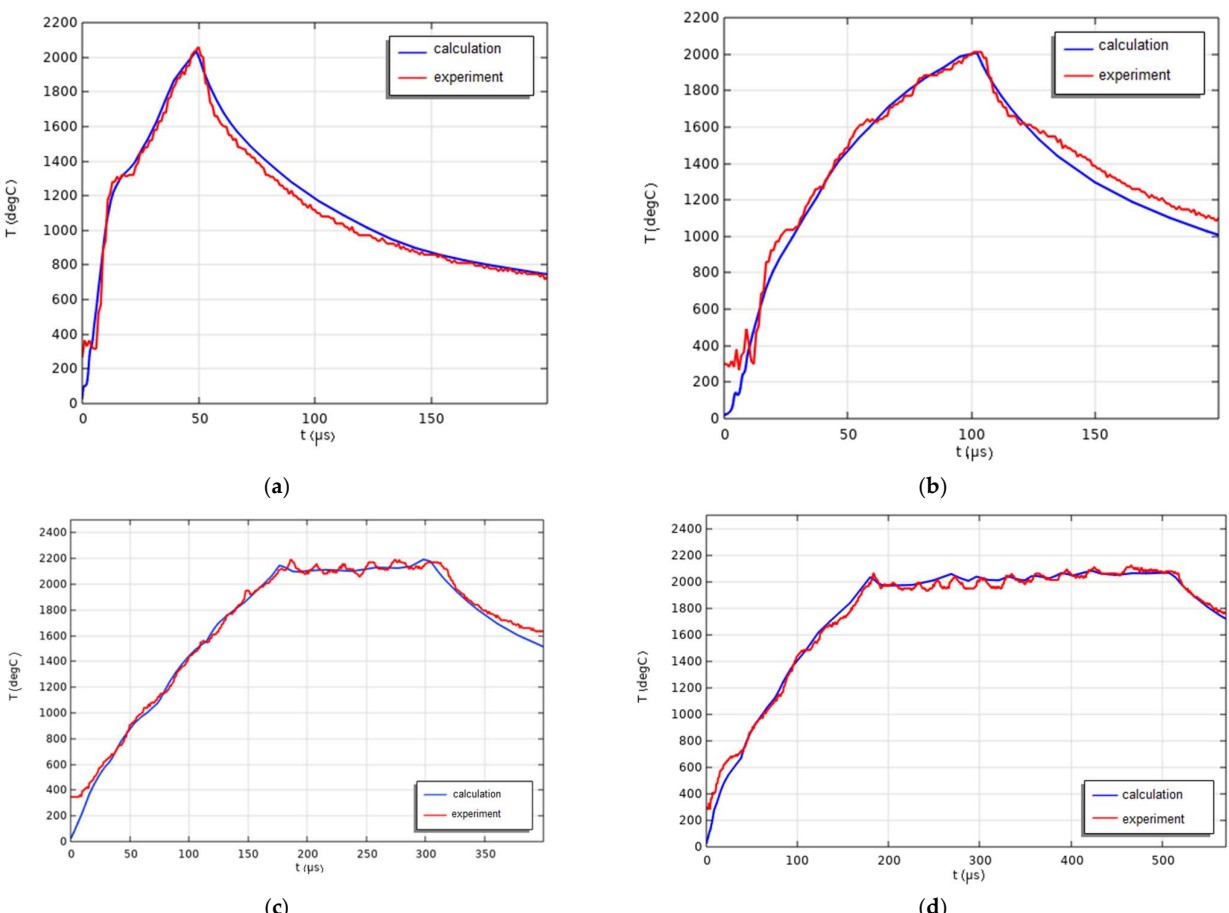

**Figure 7.** Surface temperatures for current pulse durations of 50 (**a**), 100 (**b**), 300 (**c**), and 500 µs (**d**), according to calculation and experiment.

The phase state of the steel and film–substrate system was examined on an XRD 6000 diffractometer (Shimadzu, Kyoto, Japan) in the Bragg–Brentano geometry with CuKα radiation ($\lambda$ = 1.5418 nm); the range of diffraction angles was $2\theta$ = 30–80 deg and the scan rate was 2 deg/min. The structure of the irradiated surface was analyzed on a Philips SEM-515 scanning-electron microscope (Amsterdam, The Netherlands). The elemental composition of the surface layer was determined on an EDAX ECON IV micro-analyzer (Philips, Amsterdam, Netherlands, attached to Philips SEM-515). The defect substructure and the phase state of the material were analyzed on a JEOL JEM 2100F transmission-electron microscope (Akishima, Tokyo, Japan). The surface hardness of the steel and film–substrate system at different stages of electron-ion-plasma treatment was determined on a PMT-3 hardness tester at a normal indenter load of 500 mN (LOMO, Saint Petersburg, Russia). The friction and wear coefficients of the surface layer were measured on a Tribotechnic tribometer (Tribotechnic, Clichy, France) in the pin-on-disk geometry at room temperature. The counter body was a SiC ball with a diameter of 6 mm, the track diameter was 4 mm, the rotation rate was 2.5 cm/s, the load was 5 N, and the travel distance was 1000 m. The wear volume of the surface layer was determined after track profilometry. The wear coefficient

was estimated by the formula $k = \frac{S \cdot R}{F_n \cdot n}$ mm$^3$/Nm, where $S$ is the track cross-sectional area, $n$ is the number of circles, $R$ is the circle radius, $F_n$ is the indenter load.

## 3. Analysis of Strength, Tribological Properties, and Structure

The irradiated specimens of the Nb film–AISI 5135 steel-substrate system (Nb–steel system) were tested for their surface strength (microhardness) and tribological properties (wear resistance and friction coefficient). Figure 8 shows the friction coefficient vs. time and the wear-track cross-section.

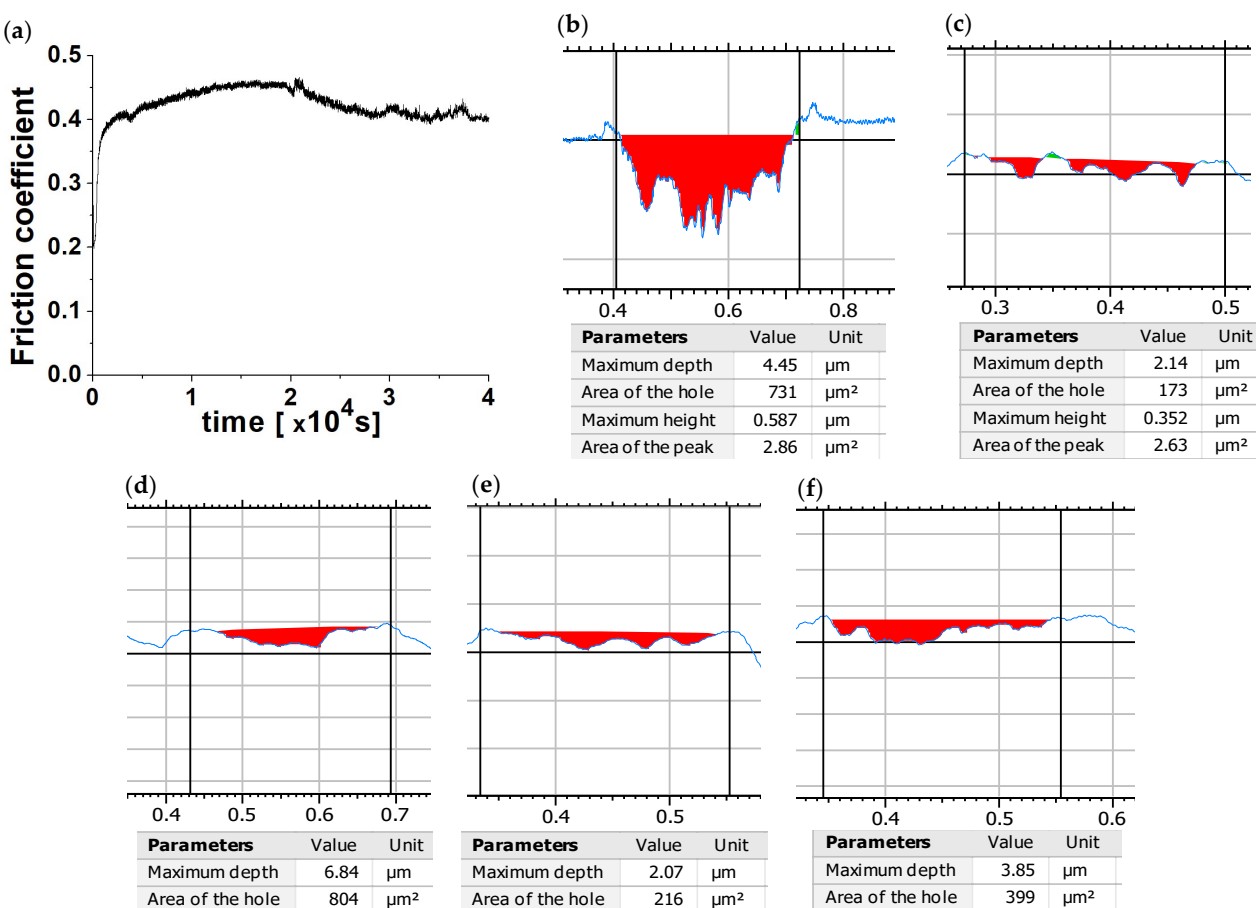

**Figure 8.** Friction coefficient vs. time (**a**) and characteristic wear-track cross-section: ((**b**) initial, (**c**) 100 μs, (**d**) 50 μs, (**e**) 500 μs, (**f**) 300 μs).

After irradiation in all four modes, the microhardness of the Nb–steel system measured 8–9 GPa, which was higher than the initial value of 3.6 GPa by a factor of 2.2–2.5 (Figure 9). Its wear rate decreased five times when the pulse duration measured 50 μs and 100 μs and three times when the temperature was maintained at about 2000 °C. For the initial sample (without coating), the microhardness was 3.6 GPa and the wear rate was $2.7 \cdot 10^{-6}$ mm$^3$/(N·m).

The surface structure and the elemental composition of the Nb–steel system before and after irradiation were examined by using scanning-electron microscopy. It can be seen in Figure 10 that microcraters formed on the surface of the system during the irradiation at pulse durations of 50–300 μs. At a pulse duration of 500 μs, no craters were detected (Figure 10e), which was probably due to the evaporation of volatile impurities from the hot specimen's surface [35]. Our X-ray spectral analysis showed that increasing the surface heating time to 300 μs decreased the Nb content in the surface layer from 97.55 at.% to 1.06 at.% (Figure 10a,d), which can be associated both with the film's immersion in the steel substrate and with the Nb's dissolution in the molten steel-surface layer.

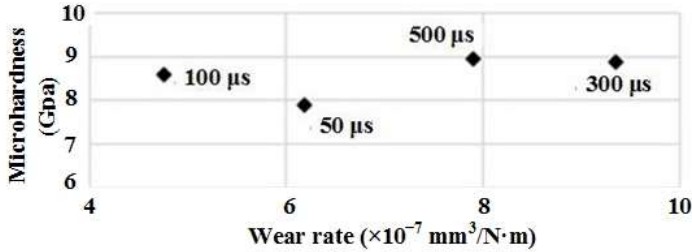

**Figure 9.** Correlation field for surface microhardness and wear rate of Nb–steel system after electron-beam irradiation.

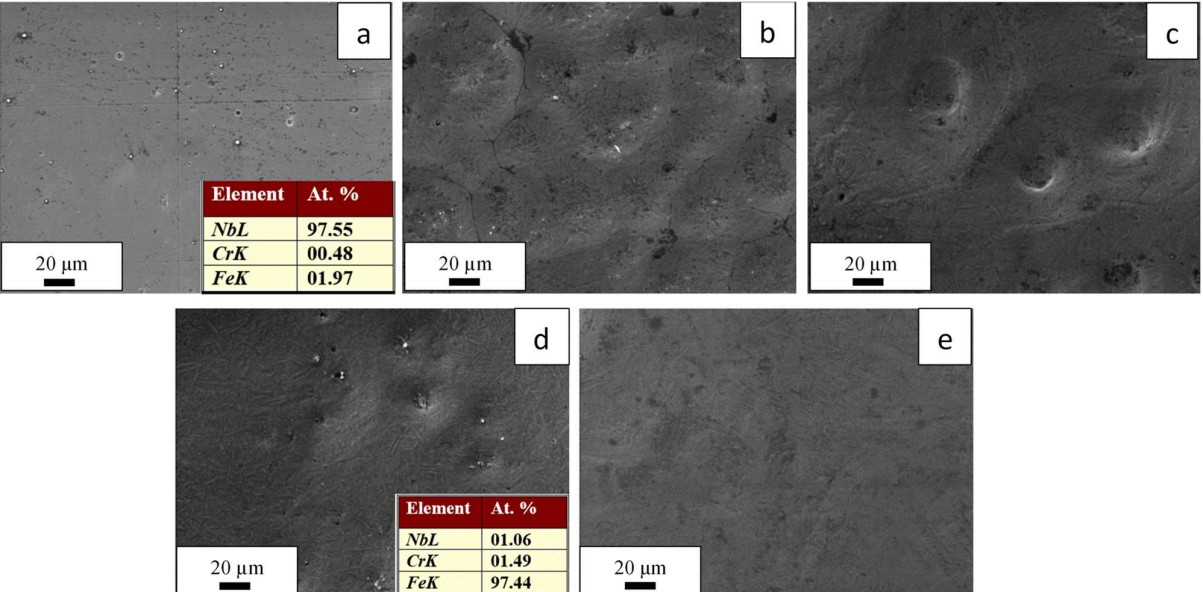

**Figure 10.** Surface structure of Nb–steel system before (**a**) and after electron-beam irradiation at pulse durations of 50 (**b**), 100 (**c**), 300 (**d**), and 500 μs (**e**).

During the electron-beam irradiation, a cellular structure of high-rate crystallization formed in the surface layer of the Nb–steel system (Figure 11), which was indicative of surface-layer melting. The average cell size increased from 0.2 μm to 0.37 μm when the pulse duration was increased (Figure 12), suggesting that the cooling rate decreased as the pulse became longer, which was confirmed by both the calculations and the real temperature measurements (Figure 7).

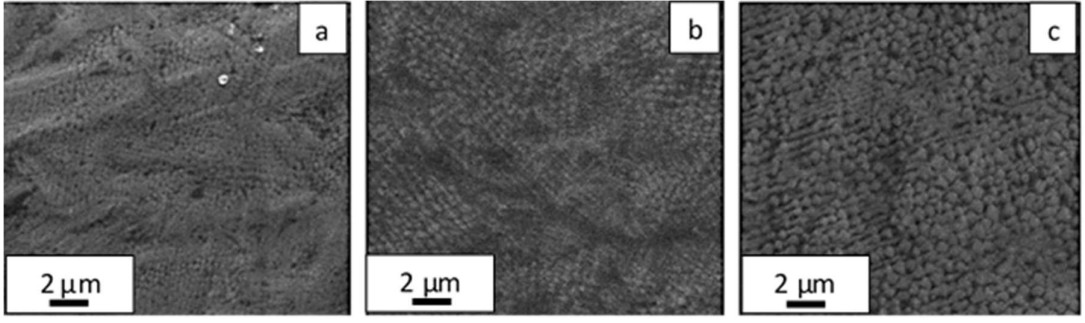

**Figure 11.** Surface structure of Nb–steel system after electron-beam irradiation for 50 (**a**), 100 (**b**), and 500 μs (**c**).

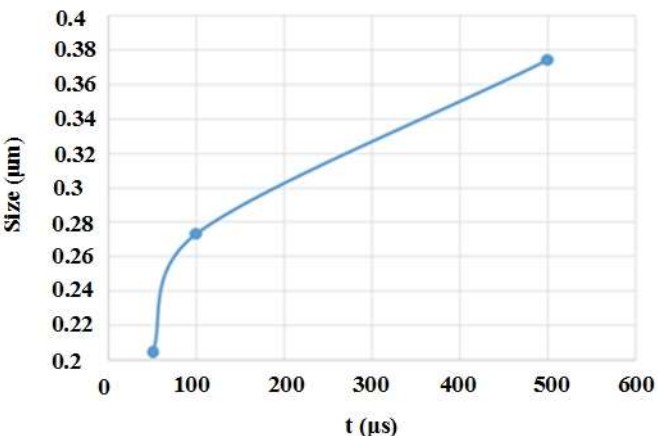

**Figure 12.** Average size of surface-crystallization cells in Nb–steel system vs. current-pulse duration.

The surface layer formed in the Nb–steel system via high-rate crystallization had a columnar structure (Figure 13). The surface-layer thickness with this columnar structure (high-rate crystallization region) agreed with the calculation data in Figure 5b, measuring about 20–25 μm at *t* = 300 μs. Our X-ray diffraction analysis showed that the Nb-surface concentration increased from 3.2 mass % (300 μs) to 4.2 mass % (500 μs) (Table 1), probably because the melt filled the surface craters and decreased the surface cooling rate. As a result, some Nb atoms migrated from the crater walls to the surface.

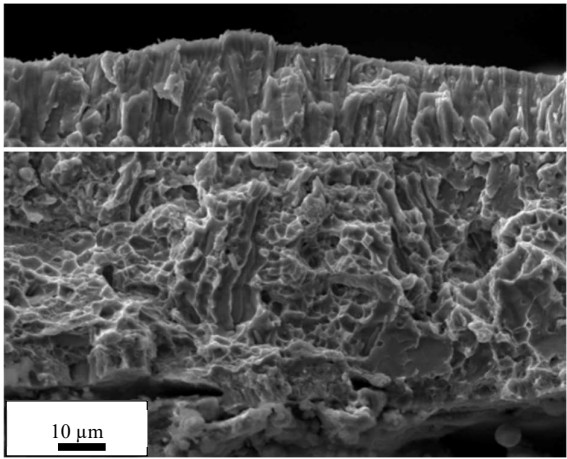

**Figure 13.** Fracture surface of Nb–steel system irradiated for 300 μs.

**Table 1.** X-ray diffraction data.

| *t*, μs | Fe, mass % | *a*(Fe), Å | *D*, nm | $\Delta d/d$, $10^{-3}$ | Nb, wt.% | *a*(Nb), Å | *D*, nm | $\Delta d/d$, $10^{-3}$ |
|---|---|---|---|---|---|---|---|---|
| 0 | 5.6 | 2.8763 | 16.7 | 0.002 | 94.4 | 3.3731 | 17.4 | 8.045 |
| 50 | 79.7 | 2.8475 | 15.6 | 1.626 | 20.3 | 3.2700 | 17.9 | 6.485 |
| 100 | 94.3 | 2.8491 | 15.4 | 1.403 | 5.7 | 3.2799 | 97.2 | 5.436 |
| 300 | 96.8 | 2.8527 | 15.2 | 1.508 | 3.2 | 3.2799 | 23.4 | 1.742 |
| 500 | 95.8 | 2.8668 | 16.0 | 1.14 | 4.2 | 3.2932 | 23.6 | 6.048 |

The data from the X-ray diffraction analysis showed that the surface layer of the Nb–steel system comprised two main phases: an α-Fe bcc solid solution and niobium (Figure 14, Table 1). During irradiation at *t* = 300 μs, the relative Nb content in its surface layer decreased from 94.4 to 3.2 wt.%. Simultaneously, the lattice parameters of Nb and

Fe changed, which was probably because the two phases diffused into each other. The formation of solid solutions was also evidenced by changes in the Nb- and Fe-lattice microdistortion ($\Delta d/d$).

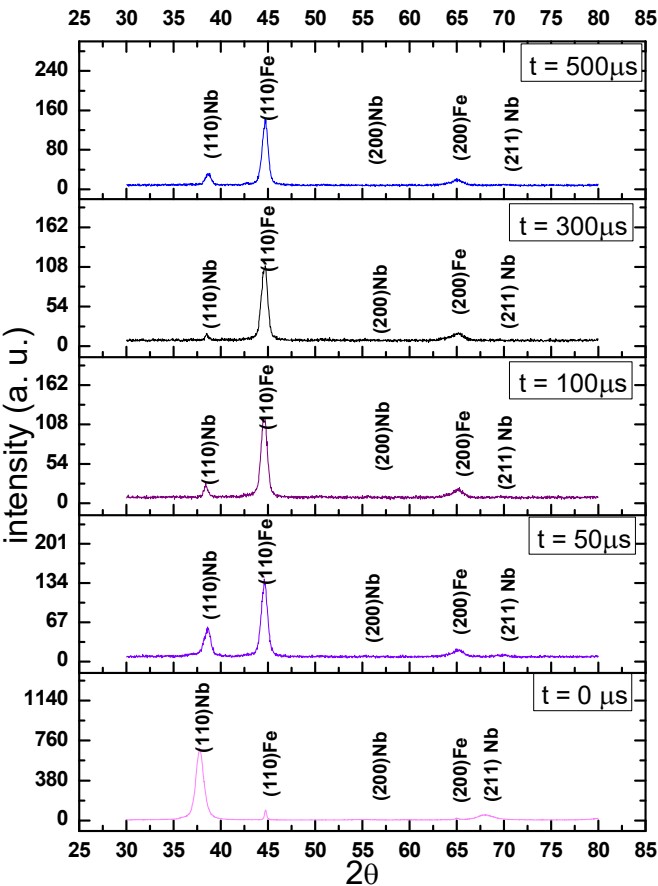

**Figure 14.** Fragments of X-ray diffraction patterns of Nb–steel system irradiated in different modes.

The data from the transmission-electron microscopy also showed that the initial structure of the AISI 5135 steel was formed by ferrite grains and perlite grains with lamellar morphologies (Figure 15).

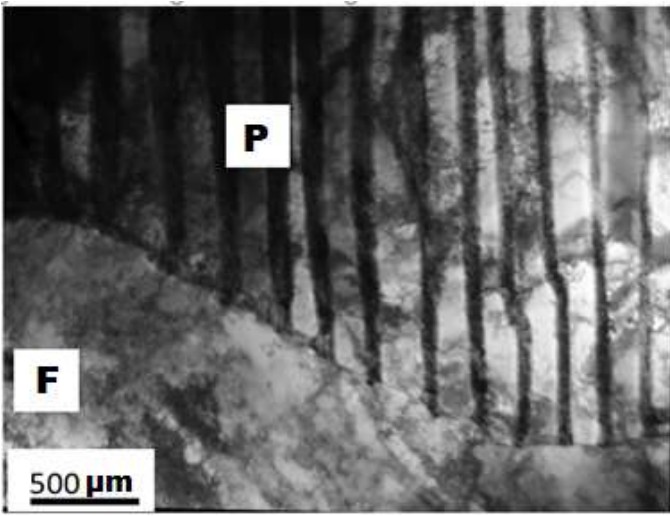

**Figure 15.** Structure of AISI 5135 steel before modification, with "P" for perlite and "F" for ferrite.

After the electron-beam irradiation of the Nb–steel system, the surface structure of the steel was substantially modified. In a surface layer of 20–30 μm, the structure was typical of high-rate quenching, containing packet martensite (Figure 16a,e) and, more rarely, lamellar martensite (Figure 16b,d). In the volume and at the boundaries of the martensite crystals, there were complex carbide $M_{23}C_6$-like nanosized particles $(Cr, Fe, Nb)_{23}C_6$ (Figure 16f).

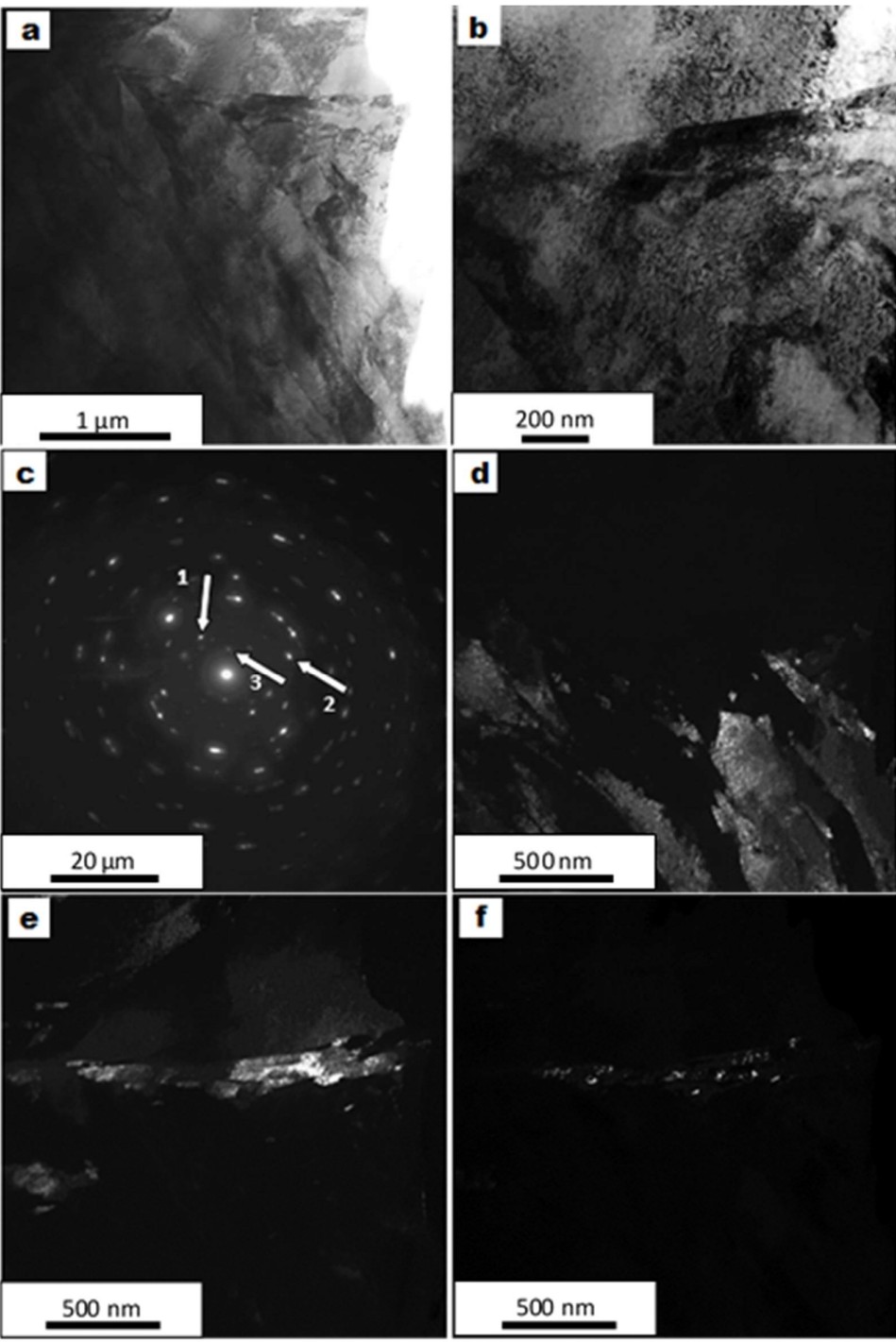

**Figure 16.** Surface structure of Nb–steel system after electron-beam irradiation at E = 24 J/cm$^2$ for 100 μs: bright fields (**a,b**), diffraction pattern (**c**), and dark fields in reflections [110]$\alpha$-Fe (**d**), [002]$\alpha$-Fe (**e**), [135]$M_{23}C_6$ (**f**), shown by arrows 1, 2, 3, respectively, in (**c**).

At a depth of 30–40 μm, the structure corresponded to the high-rate thermal transformation of pearlite (Figure 17). It comprised pearlite grains with two types of cementite

particle: extended lamellae (Figure 17a), which is characteristic of lamellar pearlite formed under quasi-equilibrium conditions; and round particles in the form of ferrite lamellae (Figure 17b), which is characteristic of the high-rate thermal transformation of cementite lamellae (dissolution and repeated precipitation).

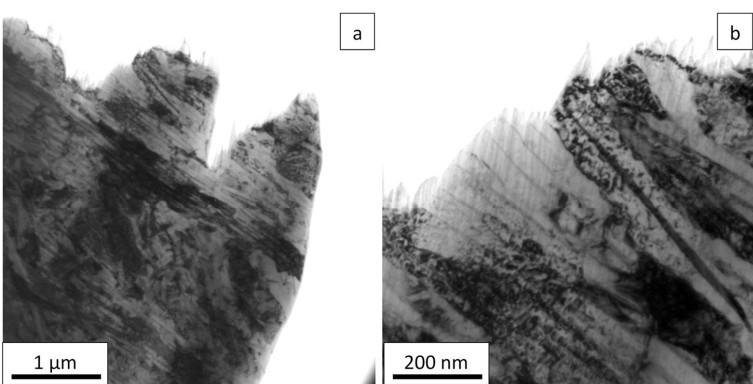

**Figure 17.** Structure of Nb–steel system 36 μm beneath its surface after irradiation at E = 24 J/cm$^2$ for 100 μm, extended lamellae (**a**), and round particles in the form of ferrite lamellae (**b**).

From the transformation of the structure and phase state of the surface layer, it was observed that the increase in the microhardness and wear resistance of the modified steel was mainly due to the formation of a hardening (martensite) structure, the precipitation of nanosized $M_{23}C_6$ particles at the martensite crystal boundaries, and pearlite structural transformation via nanosized iron-carbon precipitation in the pearlite lamellae.

At depths of 50–70 μm, the structure was close in phase state and morphology to the initial AISI 5135 steel structure. It contained ferrite and perlite grains (Figure 18).

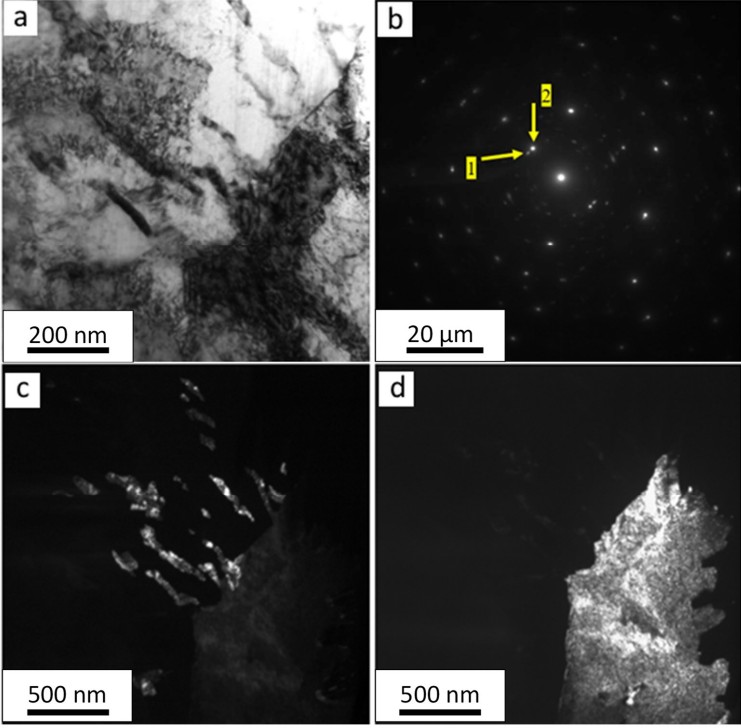

**Figure 18.** Structure of Nb–steel system 57 μm beneath its surface after irradiation at E = 24 J/cm$^2$ for 100 μs: (**a**)—bright field, (**b**)—electron-diffraction pattern, (**c**)—dark field in reflection [211]$Fe_3C$ (reflection *1* in (**b**) yellow arrow 1), (**d**)—dark field in reflection [110]$\alpha$-Fe (reflection *2* in (**b**) yellow arrow 2).

## 4. Conclusions

Structural AISI 5135 steel was exposed to surface treatment through Nb-film deposition, the surface melting of the steel, the doping of its melt with Nb via the submillisecond modulated electron-beam irradiation of the Nb film–AISI 5135 steel-substrate system (Nb–steel system), and high-rate cooling.

The advantages of steel doping with small amounts of Nb are its simplicity, highly efficient strength enhancement at a Nb content of 0.01–0.04 wt.%, high level of mechanical properties at elevated temperatures, thermal stability up to 1150 °C, and good workability and weldability, which makes it possible to decrease the weight of machines and constructions and increase their reliability and lifetime.

The irradiation was performed at pulse durations of up to 500 μs using the SOLO electron source with a grid-plasma cathode, which made it possible to vary and control the rate of energy delivery to the Nb–steel system's surface within a pulse. The temperature field formed in the Nb–steel system under intense pulsed irradiation was estimated by mathematical simulation methods, and irradiation modes were chosen to provide the near-melting conditions for the niobium and surface-melting conditions for AISI 5135 steel. The results of our study demonstrate that after irradiation in all the modes chosen, the surface microhardness of the Nb–steel system measured 8–9 GPa, which was higher than its initial value of 3.6 GPa by a factor of 2.2–2.5. The wear rate of the Nb–steel system decreased five times when the pulse duration measured 50 μs and 100 μs and three times when the temperature was maintained at about 2000 °C (t = 300 and 500 μs) due to beam-current modulation. The data from the X-ray diffraction analysis using transmission-electron microscopy showed that the increase in the microhardness and wear resistance of the modified steel was mainly due to the transformations of its surface structure and phase state: the formation of a hardening (martensite) structure, the precipitation of nanosized $M_{23}C_6$ particles at the martensite crystal boundaries, and the pearlite's structural transformation via iron-carbide precipitation in the pearlite lamellae.

Thus, our study demonstrates that such combined surface treatments of structural steels hold promise for use in industry.

**Author Contributions:** Conceptualization, V.I.S., M.S.V. and N.N.K.; methodology, M.S.V., N.N.K., T.V.K. and Y.F.I.; investigation, E.A.P., V.I.S., P.V.M., Y.F.I., N.A.P., R.A.K. and D.A.S.; resources, N.A.P.; data curation, M.S.V.; writing—original draft preparation, V.I.S., M.S.V. and Y.F.I.; writing—review and editing, V.I.S., M.S.V., N.N.K. and D.A.S. All authors have read and agreed to the published version of the manuscript.

**Funding:** The work was supported by the Russian Science Foundation (project no. 23-29-00998).

**Institutional Review Board Statement:** Not applicable.

**Informed Consent Statement:** Not applicable.

**Data Availability Statement:** Not applicable.

**Conflicts of Interest:** The authors declare no conflict of interest.

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
