# Peer review of "Steel Surface Doped with Nb via Modulated Electron-Beam Irradiation: Structure and Properties"

_coatings, doi:10.3390/coatings13061131_

Round 1
Reviewer 1 Report
The manuscript "Steel Surface Doped with Nb via Modulated Electron Beam Ir- 2 radiation: Structure and Properties" is new and original. The authors demonstrate that such combined surface treatment of structural 308 steels holds promise for use in industry. A niobium film on a AISI 5135 steel substrate was exposed to submillisecond pulsed elec- 8 tron beam irradiation with controlled energy modulation within a pulse to increase the film–sub- 9 strate adhesion. Such modulated irradiation made it possible to dope the steel surface layer with Nb 10 through film dissolution in the layer for which optimum irradiation conditions were chosen from 11 experiments and mathematical simulation.
My recommendation is "accept as is".
Author Response
Dear reviewer,
Thank you very much for your important comments and the work done.
If you have additional comments, we are ready to correct them as soon as possible.
Best regards,
group of authors
Reviewer 2 Report
This paper investigated the structure and properties of steel surface doped with Nb after modulated electron beam irradiation. The paper has done a lot of experiments, but the discussion is poor and the authors are recommended to seriously revise and resubmit the paper. The following are detailed recommendations:
1. 9 keywords are too many.
2. The format of references should be uniform.
3. Introduction is not sufficient. What is the recent status of the research on the preparation of modified layers using electron beam? Why choose Nb, why choose 5135 steel, what is the purpose and significance of this paper?
4. Line 40, how was the thickness of the Nb film (3um) determined?
5. Line 99, where does "T = 300 + 200 · Up" come from? Please give the source.
6. In the figure caption of Figure 3, "a), b), c), d)" is missing.
7. Are the graphs in Figure 3a and 3b the same?
8. Lines 126-132 and 147, each formula needs to be labeled separately. In addition all formulas should preferably be on a separate line (lines 132 and 149).
9. What is the significance of d, which is found on both sides of the equation in line 126?
10. In line 126, the density, heat capacity and thermal conductivity in Eq. should all be theoretically a function of temperature, but the authors have simplified these to a binary solid-liquid mixture, please show the justification for this in the text. In addition, the sources of the density, heat capacity and thermal conductivity used for the calculations need to be given in the text.
11. For the equation in line 127, are the authors assuming that the energy of the electron beam is deposited only on the surface of the sample? In fact the electrons have penetration depth and the energy deposition is in three dimensions. If this is a simplification for the convenience of the calculation, please clarify in the text.
12. Please keep the formula in line 129 in the same format as in line 127. And add the meaning or value of T0.
13. The temperature field calculation seems to consider only the interaction of the electron beam with the steel, how did the authors consider the effect of the Nb film in the temperature field?
14. The entire information in Figures 8a and 8b is actually the red line in Figure 5b and the blue line in Figure 7b. Figure 8 is not necessary; instead, Figures 5b and 6b and 7b lack relevant descriptions.
15. The temperature calculation curves in Figure 9 are different from those in Figures 5a, 6a, and 7a, and why?
16. The author can try to combine Figure 5, Figure 6, Figure 7 and Figure 9 into one big picture.
17. Line 205, “Its wear rate decreases 30–35 times when the pulse duration measures 50 and 100 µ s and 45–60 times when the temperature is kept at about 2000℃”, The wear rate of Nb-steel system before electron beam treatment is not given. Also, the melting point of steel is lower than 2000°C, how to kept it at about 2000°C for the wear test?
18. Please give the experimental parameters of microhardness test and friction test in the text. Please add the morphology diagram after wear and the curve of friction coefficient with time. And how is the wear rate obtained?
19. Please supplement the X-ray spectral analysis data in Figure 11.
20. Please check the corner markers in Figures 11 and 12.
21. “The average cell size increases from 0.2 to 0.37 µm with increasing the pulse duration (Fig. 13), suggesting that the cooling rate decreases as the pulse gets longer.” How was this conclusion reached? What is the basis?
22. In Figure 14, please add the graphs of 50 and 100us to support the conclusion of “thickness increases with increasing the pulse duration”.
23. What is the graph in Figure 16? surface of the sample after corrosion?
24. How to determine the ferrite and perlite grains in Figure 19?
25. Please add all experimental methods, including SEM, XRD, TEM, microhardness, friction wear, etc., in the text.
26. For all graphs, please state what graph it is before describing it.
27. There is no mention of the role of Nb throughout the text, what would be the difference if it were replaced with another element?
28. The main conclusion of the paper, "increased hardness and wear resistance", seems to be described only in lines 203-207, and no connection between the subsequent microstructural characterization and these properties is established, making the "discussion" of the whole paper seriously inadequate.
Author Response
Dear Reviewer,
We apologize for the delay in reply. It took us more than 10 days to correct the comments pointed out by the valued reviewers. Many thanks for the important comments and the work done. The article has really changed and now looks much better.
As a result, the text of the article was significantly revised, the number of keywords was corrected, a list of references was compiled, the introduction, conclusions and literature review were expanded, clarifications were made on processing methods, the equipment used and the analysis of the modified layer, graphics were corrected, equations were corrected, and much more.
If you have additional comments, we are ready to correct them as soon as possible.
Sincerely, the team of authors.
Reviewer 3 Report
1. Hardness, wear, SEM, and TEM characterization tests are not described.
2. No discussion of the results; just a report of the results obtained is presented without a scientific discussion.
Author Response
Dear reviewer,We apologize for the delay in answering.It took us more than 10 days to correct the comments indicated by the distinguished reviewers. Thank you very much for your important comments and the work done. The article has really changed and now looks much better.
As a result, the text of the article has been significantly revised, conclusions and literature review have been expanded, clarifications have been made on processing methods, the equipment used and the analysis of the modified layer and much more.
If you have additional comments, we are ready to correct them as soon as possible. Best regards, group of authors.
Reviewer 4 Report
REVIEW
for ID: 228292
Content observations
Line 39 -"... AISI 5135 with low carbon content...". In a scientific paper, the exact composition of the studied material is stated, and no qualitative assessments are made.
Fig.1 – If the applied voltage is 900 V, according to the drawing, why does the text speak (line 59) about 990 V??
Line 84 – What unit of time is "μm”?? Maybe "μs".
Lines 131-132 - Put the dimensions of the parameters listed.
Line 133 - "r0" does not appear in any equation. Why do you mention it without linking it to the logic of the exposition?
Fig. 4 – A small comment on the differences and similarities between the curves is requested. Are experiments influenced because of different aspects???
Line 143 – What is the dimension of θ? Is it taken as a percentage? How do you determine it? Or do you estimate it?
Line 154 – Give details on the thermocouple with which such high temperatures are measured. In Fig. 2 the thermocouple is positioned under the sample. The temperature of 2000 °C is achieved on the upper surface. Do I understand that you measure the temperature indirectly? In Fig. 5 declare "...surface temperature..." but measure the temperature below. You must be more detailed if you used equation (1) to estimate the temperature drop per section.
General observations
1.- You have the obligation to the readers to mention all measurement methods. For example, you omit to say how you noticed the nanometric particles of M23C6 like (Cr, Fe, Nb)23 C6 (Line 270).
2.- The actual conclusions are a summary repetition of the experiment without emphasizing the novelty of the research and defining the development directions.
3.- How was equation (1) modeled? What numerical parameters were considered? How does the model work with biphasic states as you have on the bombed surface? What parameters did you measure there? The surface is biphasic but also has mixed niobium + steel conductivity, not only liquid + solid. These aspects must be commented on, and the simplifying assumptions used must be presented.
4.- The work does not respect the structure recommended in the "Instructions for Authors". Introduction, Materials, and Methods, Results, Discussion, Conclusions.
The names of authors must also have Middle Names (especially for Russian names).
E-mail addresses are missing.
The bibliographic references do not respect the style of the magazine. Authors are listed either with their full name given explicitly or with their last name followed by their initials.
The year of publication is not bold and does not always follow the name of the publication.
The name of the publication is not italicized.
The number of pages in the journals is not given everywhere.
Also, the DOI classification of papers is not given regular
Too many self-quotes. Out of 22 titles, 12 are self-citations.
Out of 22 titles, only 11 are from the last 5 years, and of these 9 are self-citations.
Author Response
Dear reviewer,
We apologize for the delay in answering.It took us more than 10 days to correct the comments indicated by the distinguished reviewers. Thank you very much for your important comments and the work done. The article has really changed and now looks much better.
As a result, the text of the article has been significantly revised, the list of references has been drawn up, the introduction, conclusions and literature review have been expanded, clarifications have been made on processing methods, the equipment used and the analysis of the modified layer, graphics have been corrected, equations have been corrected, and much more.
If you have additional comments, we are ready to correct them as soon as possible.
Round 2
Reviewer 2 Report
The authors have carefully revised the manuscript, but a few modifications and clarifications are needed before publication.
1. Are the graphs in Figure 3a and 3b the same? And please check the figure captions of Figure 3, "a), b), c), d)".
2. In formula (2), can not be written in the form of
, also,
is wrong.
3. On page 6, "At an electron beam energy of <25 keV, the energy source in the mathematical model can be taken to act at the surface" is not well founded. For the calculation of the electron beam range, refer to the formula in doi: 10.3 390/coatings11080912:
4. On page 9, it is unclear about the object with the hardness of “5 GPa”, either the 5135 steel or the Nb-steel system before electron-beam treatment. It is also unclear about the comparison object of the decrease in wear rate, either the 5135 steel or the Nb-steel system before electron-beam treatment? Besides, please add the wear rate value of the initial sample. Supplement the curves of friction coefficient with time and the morphology after wear for each sample before and after electron beam treatment in Figure 8. The results of hardness and wear rate cannot be the result of one measurement, but are the average of several measurements including error bars.
5. On page 9, "Our X-ray spectral analysis shows that increasing the surface heating time to 300 µs decreases the Nb concentration in the surface layer from 91.0 to 1.95 at%", dose this conclusion come from XRD? If it is from XRD, only "wt.%" is shown in Table 1, please unify the units. If not, please add the corresponding graphs of the results.
6. Please check the symbols in Figure 10 and 11.
7. On page 10, "The average cell size increases from 0.2 to 0.37 µm with increasing the pulse duration (Fig. 11), suggesting that the cooling rate decreases as the pulse gets longer", please discuss this with the calculated cooling rate in Figure 7. The decrease in cooling rate may be the reason for the increase in cell size, not the result, so "suggesting that" cannot be used.
8. The last paragraph on page 11 is Figure 15 rather than Figure 16. Besides, delete "also" in the sentence.
9. The first sentence of the Abstract, "A niobium film on an AISI 5135 steel substrate was exposed to submillisecond pulsed electron beam irradiation with controlled energy modulation within a pulse to increase the film-substrate adhesion", seems to focus on "the difference between the Nb- steel system before and after electron beam treatment". However, there is no study related to film-substrate adhesion in the paper, and only the difference between Nb-steel system after electron beam treatment and 5135 steel is analyzes in the Results and Discussion section. Therefore, please clarify the purpose and significance of the study in the Abstract and these contents also need to state at the end of the Introduction.
10. The first two paragraphs of Conclusion are the research background rather than the conclusion.
Author Response
Thank you dear reviewer! We have tried to take into account all your comments and thanks to you the article has become significantly better.
All changes are highlighted in the text of the article.
P.S.: Figure 8a shows the characteristic dependence of the friction coefficient on time, with different processing modes, only the time required for the wear of the image changed, from 50 m for the initial to 1000 after processing.
Kind regards, group of authors
Reviewer 4 Report
In the modified form the paper deserve to be publishAuthor Response
Thank you dear reviewer! The article has become significantly better.
Kind regards, group of authors
Round 3
Reviewer 2 Report
1. Figure 3a and 3b are still the same.
Author Response
Dear reviewer! Thank you for your care and attention to detail! You are absolutely right. Since many authors are working on the work, at one of the stages of work we once again crept in such an annoying typo. We bring our changes and insert the correct image. Our team is very grateful for your help!